# Serosal Adhesion Ex Vivo of Hydrogels Prepared from Apple Pectin Cross-Linked with Fe^3+^ Ions

**DOI:** 10.3390/ijms24021248

**Published:** 2023-01-08

**Authors:** Sergey Popov, Nikita Paderin, Elizaveta Chistiakova, Dmitry Ptashkin

**Affiliations:** Institute of Physiology of Federal Research Centre “Komi Science Centre of the Urals Branch of the Russian Academy of Sciences”, 50, Pervomaiskaya Str., 167982 Syktyvkar, Russia

**Keywords:** apple pectin, Fe^3+^ ions, hydrogel, serosa, bioadhesion, mechanical properties, scanning electron microscopy, adhesion medium

## Abstract

The study aims to investigate the adhesion of a hydrogel made of cross-linked low-methyl esterified pectin to rat intestinal serosa ex vivo. The adhesivity of the FeP hydrogel, which was cross-linked by Fe^3+^ cations, exceeded that of hydrogels cross-linked by Ca^2+^, Zn^2+^, and Al^3+^ cations. The concentration of the cross-linking cation failed to influence the adhesion of the pectin hydrogel to the serosa. The mechanical properties and surface microrelief of the pectin hydrogel were influenced by the type and concentration of the cross-linking cations. Fe^3+^ cations form a harder and more elastic gel than Ca^2+^ cations. Scanning electron microscopy analysis revealed the characteristic surface pattern of FeP hydrogel and its denser internal structure compared to Ca^2+^ cross-linked hydrogel. The effect of the salt composition of the adhesion medium was shown since the FeP hydrogel’s adhesion to the serosa was lower in physiological solutions than in water, and adhesion in Hanks’ solution was higher than in phosphate buffered saline. Serum proteins and peritoneal leukocytes did not interfere with the serosal adhesion of the FeP hydrogel. Pre-incubation in Hanks’ solution for 24 h significantly reduced the adhesion of the FeP hydrogel to the serosa, regardless of the pH of the incubation. Thus, serosal adhesion combined with excellent stability and mechanical properties in physiological environments appeared to be advantages of the FeP hydrogel, demonstrating it to be a promising bioadhesive for tissue engineering.

## 1. Introduction

Pectin is a plant polysaccharide that is widely used in several scientific fields because of its availability, safety, and functionality [1,2]. Linear and branched regions alternate in the pectin macromolecule and have a backbone that is made up of partially methyl-esterified 1,4-linked D-galacturonic acid (GalA) residues. Pectins may be classified into high- (HM) and low-methyl esterified (LM) pectins based on the percentage of GalA units esterified with methanol (degree of methyl esterification). The degree of methyl esterification of HM and LM pectin is greater and less than 50%, respectively. Arabinans, galactans, and/or arabinogalactans predominantly make up the side chains of the branched regions of pectin [3]. HM pectins form physical gels at pH < 3.5 in the presence of co-solutes, and LM pectins gel in the presence of multivalent cations in a wide pH range [4]. In the pharmaceutical industry, pectin is used as a carrier for colonic drug delivery systems because of its resistance in the human stomach and small intestine and degradation in the large bowel by colonic bacteria [5]. In addition, the mucoadhesiveness of pectin is being intensively studied in the development of targeted drug delivery. Previous research has shown the mucoadhesive properties of pectin, which largely depend on its characteristics, i.e., the degree of esterification and molecular weight. Physical entanglement of pectin and mucin because of chain association by ionic and hydrogen bonding has been suggested [6].

Biodegradability, biocompatibility regarding different tissues, drug loading/releasing capacity, and tunable mechanical properties contribute to the application of pectin gels as promising materials for biomedicine [7]. The interaction of pectin with the serous membrane lining the surface of the human body cavities, as well as covering the internal organs, must be taken into account when considering pectin biomaterials for surgery, tissue engineering scaffolds, and biosensors. Recently, Servais et al. [8] suggested that the bioadhesive properties of pectin are not limited to mucoadhesivity but may be more general. It was found that pectin may also be effective at targeting the glycocalyx of visceral mesothelium, including the mesothelium of the lung, liver, bowel, and heart [8]. Several potential benefits of applying pectin as a mesothelial bioadhesive have been demonstrated. In particular, pectin was proposed as a potential patch-sealant for pleural air leaks [9,10,11] and a sealant for anastomotic healing after intestinal surgery [12]. Pectin-based compounds are suggested to provide a scaffold for mesothelial regeneration, facilitate the delivery of drugs or growth factors to the mesothelial surface, and inhibit post-surgical serous adhesions [8].

According to [8], the pectin-mesothelial interaction was stronger using HM pectin compared to LM pectin, therefore serosal adhesivity was studied mainly for HM pectin. Here, we suggest that an ionotropic hydrogel based on LM pectin might be of interest as a serosal adhesive for the following reasons. Pectin serosal adhesion appears to be the result of the entanglement or interpenetration of pectin chains with mesopolysaccharides covering the mesothelium, analogous to the branched-chain entanglement that occurs with the mucoadhesion of pectin, and water movement was shown to provide the motive force for the rapid pectin chain entanglement [13]. It can, therefore, be expected that LM pectin hydrogel will provide a source of hydration that facilitates chain interactions since the water content of pectin hydrogel exceeds 90%. Cations, which cross-link the pectin network, may produce an additional electrostatic mechanism of interaction between the pectin and the glycocalyx. In addition, cross-linking cations can change the physicochemical properties and surface microrelief of the pectin hydrogel, which are important for bioadhesivity. Calcium ions are mainly used as a cross-linker to obtain ionotropic LM pectin hydrogels. Other divalent (Zn^2+^, Mg^2+^) [14,15,16,17] and trivalent (Fe^3+^, Al^3+^) [18,19,20,21] cations are increasingly being considered as an alternative to calcium ions in order to improve the physicochemical and functional properties of calcium pectin hydrogels. However, the effect of the cross-linking cation on the adhesiveness of the pectin hydrogel to the serosa has not been studied before.

In this paper, we investigated the adhesion to rat small bowel serosa ex vivo of hydrogel prepared from apple pectin cross-linked by Fe^3+^ cations. The study compares the effects on serosal adhesion of cross-linking cations, the mechanical properties of hydrogel, and the adhesion conditions.

## 2. Results

### 2.1. Effect of the Type and Concentration of a Cross-Linking Cation on the Adhesivity of Pectin Hydrogel

The screening for adhesivity of hydrogels prepared from a 4% solution of apple pectin using ionotropic gelling with Ca^2+^, Zn^2+^, Fe^3+^, and Al^3+^ cations revealed the Fe^3+^ cross-linked pectin (FeP) hydrogel for mesothelial adhesion. The adhesion strength of hydrogels with Fe^3+^, Zn^2+^, and Al^3+^ cross-links was higher by 25, 18, and 18% than that of a Ca^2+^ cross-linked hydrogel (Figure 1). The work of adhesion for the FeP hydrogel was the highest when compared with the pectin hydrogels cross-linked by other cations, exceeding the work of adhesion for the Ca^2+^ cross-linked (CaP) hydrogel by about two times. 

The hydrogel probe was descended onto the serosa and maintained with a constant compression force in this test (Figure 2i). The deadhesion phase of the assay was the detachment of the hydrogel probe (Figure 2ii) with a peak force corresponding to the adhesion strength and a debonding curve (Figure 2iii), during which the adhesion force decreased because of the gradual breaking of bonds between the hydrogel and the serosa. The protracted segment of the debonding curve (Figure 2, arrows) reflected the entanglement of the polysaccharides of the interacting surfaces. Adhesion became weaker with several repeated attachment-detachment contacts of the FeP hydrogel with the serosa. The adhesion strength decreased by 3% and 6% after the second and fifth cycles of adhesion, compared to the first cycle. Interestingly, the protracted segment of the debonding curve was not observed on the adhesion profile after five adhesion-deadhesion cycles of the FeP hydrogel (Figure 2B). The adhesion did not change during five successive cycles of attachment-detachment of the CaP hydrogel to the serosa.

The concentration of the cross-linking cation failed to influence the adhesion of the pectin hydrogel to the serosa. The concentrations of cross-binding ions in this study corresponded to three ratios of the number of cations and free carboxyl groups of GalA in the pectin (stoichiometric ratios R = 0.5, 1.0, and 2.0). Calcium ion concentrations were 31, 62, and 124 mM to reach R = 0.5, 1.0, and 2.0, respectively, according to the formula R = 2(Ca^2+^)/(COO^−^). The iron (III) ion concentrations were 21, 42, and 84 mM to reach R = 0.5, 1.0, and 2.0, respectively, according to the formula R = 3(Fe^2+^)/(COO^−^).

Thus, the pectin hydrogel (FeP) cross-linked by Fe^3+^ (124 mM) was chosen for further investigation, while the Ca^2+^ cross-linked hydrogel (CaP) was used for comparison purposes. 

### 2.2. Mechanical Properties and Surface Microrelief of FeP and CaP Hydrogels

An increase in the concentration of the cross-linking cation significantly changed the mechanical properties of the pectin hydrogels (Table 1). The hardness of the FeP hydrogel obtained at 42 mM of Fe^3+^ was 24% higher than that of the FeP hydrogels obtained at 21 and 84 mM of Fe^3+^. The Young modulus and adhesiveness of the FeP hydrogel increased with an increase in the concentration of the cross-linking cation. On the contrary, the brittleness of the FeP hydrogel decreased by 38–44% with an increase in the concentration of Fe^3+^. The hardness of the FeP hydrogel exceeded that of the CaP hydrogel by 4.5, 4.2, and 3.0 times at cation concentrations corresponding to 31, 62, and 124 mM, respectively. The Young modulus of the FeP hydrogel exceeded that of the CaP hydrogel by 2.0, 6.4, and 4.2 times at corresponding cation concentrations, respectively. 

Scanning electron micrographs illustrating the surface morphology of the FeP and CaP hydrogels, are presented in Figure 3. The FeP hydrogel surface had a wrinkly microrelief with protruding fiber-like formations 1–2 µm long (Figure 3, arrows). The CaP hydrogel surface had a smoother microrelief with visible pits 1–2 µm diameter (Figure 3, dotted circle).

Scanning electron micrographs of a cross section of the FeP hydrogel showed densely packed hydrogel layers with 100–120 µm pores (Figure 4A–C). The CaP hydrogel had a looser internal structure with pores larger than 200 µm (Figure 4D–F). EDS analysis revealed similar contents of carbon (40–47 wt%) and oxygen (43–52 wt%) in the FeP and CaP hydrogels. The Fe^3+^ and Ca^2+^ ion content was 9.2 ± 3.6 and 8.5 ± 2.1 wt% in the FeP and CaP hydrogels, respectively.

Thus, the mechanical properties and surface microrelief of pectin hydrogel were influenced by the type of cross-linking cations. Fe^3+^ cations form a harder and more elastic hydrogel than Ca^2+^ cations. SEM analysis revealed the characteristic surface pattern of the FeP hydrogel and its denser internal structure compared to the CaP hydrogel. 

### 2.3. Effect of the Adhesion Conditions on the Adhesivity of the FeP Hydrogel

The effect of the composition of the adhesion medium on the adhesivity of the FeP hydrogel is shown in Figure 5. In this experiment, 0.1 mL of distilled water (DW), phosphate buffered saline (PBS), Hanks’ solution, Hanks’ solution supplemented with 10% fetal bovine serum (FBS), and Hanks’ solution supplemented with 10% FBS and peritoneal leukocytes were placed between the serosa and the hydrogel during the adhesion test so that the liquid covered the entire area of the contact surfaces. The adhesion strength of the FeP hydrogel to the serosa was 37% lower in PBS than in distilled water. The decrease in adhesion strength was 10–20% in Hanks’ solutions compared to distilled water and did not depend on the presence of serum proteins or peritoneal cells. 

An increase in both the force and time of compression of the FeP hydrogel to the serosa increased the adhesion strength and work of adhesion (Figure 6). Depending on the compression force, the adhesion strength was greater by 6–18 and 22–30% when the FeP hydrogel was compressed to the serosa for 60 and 120 s than for 20 s. When the FeP hydrogel was compressed to the serosa with forces of 50 and 100 mN, rather than 25 mN, the adhesion strength increased by 10–14 and 26–41%, respectively, depending on the compression time.

Thus, the effect of the salt composition of the adhesion medium was shown since the FeP hydrogel’s adhesion to the serosa was lower in physiological solutions than in water, and adhesion in Hanks’ solution was higher than in PBS. Serum proteins and peritoneal cells did not interfere with the serosal adhesion of the FeP hydrogel. The force and time of the compression of the hydrogel to the serous tissue were directly related to adhesion.

### 2.4. Stability of FeP Hydrogels Incubated at Different pH

The mechanical properties of the FeP hydrogel and the release of GalA into the incubation medium were measured during the incubation of the FeP hydrogel in Hanks’ solution at pH 7.4, 5.0, and 8.0 for 24 h. The mechanical properties of the FeP hydrogel did not change significantly upon incubation in Hanks’ solution with pH 7.4 for 24 h (Figure 7). GalA was gradually released into the incubation medium (Figure 8A), and the pH of the incubation medium decreased from 7.4 to 6.8 after 3 h of incubation with the FeP hydrogel and remained at this level for 24 h (Figure 8B). The mechanical properties of the FeP hydrogel did not change significantly upon incubation in Hanks’ solution with pH 5.0 for 24 h (Figure 7A). 

GalA was released significantly less during incubation at pH 5.0 than at pH 7.4 (Figure 8A). The hardness of the FeP hydrogel decreased by two and four times after 3 and 6 h of incubation in Hanks’ solution with pH 8.0 and remained at this level for 24 h (Figure 7A). This decrease in hardness was accompanied by a significant increase in the adhesiveness of the FeP hydrogel (Figure 7B). GalA was intensively released from the FeP hydrogel upon incubation at pH 8.0 (Figure 8A). The pH decreased to 7.5 during the first 6 h of incubation of the FeP hydrogel in Hanks’ solution from an initial pH of 8.0 and then recovered to 8.0 after 24 h of incubation (Figure 8B).

Iron ions were not released from the FeP hydrogel upon incubation at pH 7.4 and 8.0. However, 1.5–2.0 mM iron ions were found in the incubation medium when the FeP hydrogel was incubated at pH 5.0 (Figure 9). Considering that GalA was not released during the incubation of the FeP hydrogel at pH 5.0, the data demonstrate the release of iron ions, which are not involved in the cross-linking of pectin chains.

Thus, the good stability of the FeP hydrogel in acidic and physiological environments was shown. In an alkaline environment, degradation of the pectin hydrogel occurred, as evidenced by a decrease in the gel strength and the release of GalA from it.

### 2.5. Serosal Adhesion of FeP Hydrogels Pre-Incubated at Different pH

Pre-incubation in Hank’s solution for 24 h significantly reduced the adhesion of the FeP hydrogel to the serosa, regardless of the pH of the incubation (Figure 10). The adhesion strength of the FeP hydrogel pre-incubated at pH 7.4, 5.0, and 8.0 for 24 h was 2.2, 1.8, and 1.8 times lower than the initial value, respectively (Figure 10A). The decrease in adhesion strength was accompanied by a decrease in the work of adhesion after incubation of the FeP hydrogel at pH 7.4 and 5.0. However, the work of adhesion decreased after 3 h and then recovered to the initial level after 24 h of incubation at pH 8.0 (Figure 10B). 

The protraction of the debonding curve was found in the adhesion profile of the FeP hydrogel pre-incubated at pH 8.0 (Figure 11D, arrow). It should be noted that the protracted segment of the debonding line was less pronounced after incubation of the FeP hydrogel at pH 5.0 (Figure 11B). It was calculated that the part of the adhesion work corresponding to the protracted segment of the adhesion curve was 15.6 ± 5.0% of the total adhesion work of the non-treated FeP hydrogel. The part of the adhesion work corresponding to the protracted segment of the adhesion curve was 23.9 ± 7.2, 7.3 ± 2.7, and 42.3 ± 16.5% of the total adhesion work of the initial FeP hydrogel.

Thus, a decrease in the adhesiveness of the FeP hydrogel after its incubation was revealed, regardless of the pH of the incubation solution. It was found that the protracted segment of the debonding curve decreases and increases in samples of FeP hydrogel preincubated in acidic and alkaline conditions, respectively, in comparison with hydrogel preincubation in physiological conditions.

### 2.6. Subcutaneous Implantation of Pectin Hydrogels in Rats

For the in vivo biocompatibility studies, hydrogel samples measuring 10 × 10 × 5 mm (width, length, and height) and weighing 0.2–0.4 g were implanted subcutaneously in laboratory Wistar rats. The hydrogels were removed 1 and 7 days after implantation and weighed. Within 7 days, the animals were weighed and observed for behavior. It was established that the subcutaneous implantation of the FeP and CaP hydrogels did not affect body weight changes or postoperative behavior of rats in comparison with sham-operated animals. The body weight of all animals was reduced after 1–7 days of surgery by an average of 10–11 and 6–9 g compared with their original weight. The total number of leukocytes, and the content of neutrophils and mononuclear cells in the blood of rats implanted with the pectin hydrogels did not differ from those in sham-operated animals. Tumor necrosis factor-a (TNF-a) levels in the blood of rats implanted with the CaP hydrogel did not change after 7 days. TNF-a levels in the blood of rats implanted with the FeP hydrogel were 1.8 times higher than in sham-operated rats (3.5 ± 1.9 and 1.8 ± 1.7 ng/mL, respectively) after 1 day of implantation and fell to below the measurement threshold by the 7th day. 

The CaP hydrogel kept its shape 1 day after implantation and transformed into an amorphous hydrogel clot 7 days after implantation (Figure 12A,B). One day after implantation, the FeP hydrogel softened and lost its original shape (Figure 12C,D).

The gel material removed from the implantation site was treated with the fluorescent dye 4’,6-diamidino-2-phenylindole (DAPI) to assess cell adhesion by counting the number of cells on the implanted hydrogels. The implanted hydrogel material was found to be infiltrated by cells, with the degree of cellular infiltration increasing with increasing implantation time (Figure 13). After 1 day of implantation, the number of cells on the CaP and FeP hydrogel implants was 971 ± 405 and 491 ± 280 cells/mm^2^, respectively (Figure 13A,C). After 7 days of implantation, the number of adherent cells increased to 3686 ± 991 cells/mm^2^ for the CaP hydrogel and 1124 ± 315 cells/mm^2^ for the FeP hydrogel (Figure 13B,D). The data obtained indicated a higher biocompatibility of the FeP hydrogel than the CaP hydrogel.

## 3. Discussion

Hydrogels prepared from a 4% solution of LM apple pectin using ionotropic gelling with Ca^2+^, Zn^2+^, Fe^3+^, and Al^3+^ cations were found to adhere to the serosa of rat small intestine. The FeP hydrogel demonstrated the highest strength and work of adhesion among the hydrogels obtained and, therefore, was chosen for further investigation. The adhesion strength of the FeP hydrogel at a contact area of 28 mm^2^ ranged from 50 to 140 mN, which corresponded to 1.6–6.4 kPa. HM citrus pectin film has previously been shown to adhere to porcine small bowel serosa with a strength of 2.8 kPa [12]. Therefore, our assumption was confirmed that the LM pectin hydrogel could be interesting as a serosal adhesive. It should be noted that the adhesion of the LM pectin hydrogel to the serosa was weaker than the interadhesion of two pectin films, which was in the range of 28–169 kPa according to [13]. This is probably due to the fact that the adhesiveness of serosa mesopolysaccharides appears to be lower than the adhesiveness of pectin. The structure of serous mesopolysaccharides remains poorly understood, although it is clear that they provide the slippery, low-adhesion surface of visceral organs [22].

As shown by the initial screening, the type of cross-linking cation affects the adhesiveness of the pectin hydrogel. The type of cross-linking cation is well known to be a significant determinant of pectin gel stability [16,23], drug-releasing behavior [15,20], biocompatibility [24], and other functional properties [25]. Calcium ions are mainly used as a cross-linker to obtain ionotropic pectin hydrogels, however, Zn^2+^ and Al^3+^ ion cross-linking provides better control of stability and drug release rate than Ca^2+^ [20,21,23]. In the present study, the bioadhesion of hydrogels with Zn^2+^, and Al^3+^ cross-links was stronger than that of a Ca^2+^ cross-linked hydrogel. However, we did not further consider the adhesiveness of these hydrogels due to their low biocompatibility. It was previously shown that Zn^2+^ cross-linked hydrogel had a cytotoxic effect on leukocytes and fibroblasts, and Al^3+^ cross-linked hydrogel enhanced the production of the inflammatory cytokine IL-1β [24]. 

An unexpected finding was that the concentration of the cross-linking Fe^3+^ failed to influence the adhesion of the FeP hydrogel to the serosa. According to previous results and the generally accepted model of pectin gelation, an increase in the concentration of the cross-linking cation increased the hardness of the pectin hydrogel. LM pectin forms hydrogel in the presence of divalent and trivalent cations by the so-called “egg box” mechanism [26,27]. The “egg-box” model for junction zone formation by divalent cations, such as Ca^2+^, involves two ionic links between the free carboxylic acid groups (COO−) of two neighboring pectin chains [4]. In addition to ionic bridges, egg-boxes are stabilized by hydrogen bonds and van der Waals interactions. The trivalent cations, such as Fe^3+^ are suggested to form a stronger three-dimensional network due to the possibility of forming an additional ionic bond between pectin chains (Figure 14). 

Cross-linking has been previously shown to reduce pectin mucoadhesion, probably by reducing the number of chains available for entanglement with the adhesion substrate [28]. A cross-linking cation concentration corresponding to the stoichiometric ratio (R) equal to 0.5 indicates that the available COO− charges are screened by cations partially, whereas all pectin charges are occupied by cations at R 1.0 and over [29]. Therefore, it could be expected that the serosal adhesion of the pectin hydrogel would decrease with an increase in the concentration of the cross-binding cation. Our results suggest that the bioadhesiveness of LM pectin hydrogel weakly depends on the degree of cross-linking of pectin chains. Perhaps the surface morphology of the hydrogel was important, since the FeP hydrogel surface, which had a microrelief with protruding fiber-like formations, differed from the surface of the CaP hydrogel with pits.

The salt composition of the adhesion medium influenced the serosal adhesion of the FeP hydrogel in the following order: PBS < Hanks’ solution < distilled water. These data may indicate that cations in PBS and Hanks’ solutions affect adhesion because they additionally bind the polysaccharide chains of the hydrogel to each other and/or the mesopolysaccharide chains to each other, and this reduces their availability for mutual entanglement. In distilled water, the polysaccharide chains of the contacting surfaces are probably less susceptible to mutual entanglement. At the same time, calcium and magnesium cations from the composition of Hanks’ solution may promote adhesion compared to PBS, which does not contain multivalent cations, because of the formation of additional cross-links between pectin and mesopolysaccharides.

Protein adsorption and leukocyte adhesion on the surface are one of the first events after biomaterial implantation [30,31]. We have previously shown that pectin hydrogels adsorb serum proteins, and adhere and activate leukocytes [24,32,33]. In the present study, serum proteins and peritoneal cells did not interfere with the serosal adhesion, suggesting different binding points. The independence of adhesion to serosa from the presence of serum proteins and leukocytes appears to be important for the practical application of LM pectin hydrogel, as it allows its use in inflammatory conditions without loss of functionality. Furthermore, pectin hydrogel has the advantage of being stable at acidic and neutral pH. However, regardless of the incubation pH, serosal adhesion of FeP hydrogel was significantly reduced after 24 h incubation in Hank’s solution.

The limitations of this study are related to the lack of data on the potential mechanism of adhesion of LM pectin hydrogel to serosa. Protraction of the debonding curve of FeP adhesion to serosa indirectly confirms the entanglement of mesopolysaccharides chains with pectin polysaccharide chains. However, the polymer chains that are involved in gel network formation are hardly accessible for contact with the adhesion substrate. As a result, the galacturonan backbone is unlikely to provide pectin hydrogel adhesion to the serosa. The pectin macromolecule contains branched regions with side chains of neutral monosaccharide residues [3]. Neutral side chains of pectin do not participate in gel cross-linking and, therefore, can interpenetrate with serosal mesopolysaccharides. Apple pectin (AU701) used in the study consisted of 86.5, 2.8, 2.3, and 0.6 wt% of GalA, xylose (Xyl), galactose (Gal), and arabinose (Ara), respectively. The percentage of the homogalacturonanic (HG) region calculated as (GalA–Rha) in AU701 was 85.2, indicating that HG regions were dominant in the pectin used. The contribution of rhamnogalacturonan-I (RG-I) (Rha/GalA) to the pectin’s structure and the branching degree of RG-I ((Gal+Ara)/Rha) were 0.02 and 2.2, respectively. The Xyl/GalA ratio used for estimation of the xylogalacturonanic (XGA) domain was 0.03. The data obtained indicate that pectin side chains, including Ara, Gal, and Xyl residues, can participate in adhesive interactions with serosa. However, the exact molecular mechanism of pectin gel adhesion to serosa remains unclear.

In conclusion, a hydrogel made of cross-linked LM pectin adhered to rat intestinal serosa ex vivo with a strength that depended on the type of cross-linking cation. The adhesivity of the FeP hydrogel, which was cross-linked by Fe^3+^ cations, exceeded that of hydrogels cross-linked by Ca^2+^, Zn^2+^, and Al^3+^ cations. Serosal adhesion of the FeP hydrogel seemed to increase in the adhesion medium containing multivalent cations and decreased after incubation of the hydrogel in Hanks’ solution at pH 5.0, 7.4, and 8.0 for 24 h. Serosal adhesion combined with excellent stability and mechanical properties in physiological environments appeared to be advantages of the FeP hydrogel, demonstrating it to be a promising bioadhesive for tissue engineering.

## 4. Materials and Methods

### 4.1. Preparation of Pectin Hydrogels

Apple pectin (AU701, Herbstreith and Fox (Nuremberg, Germany)) used in the study consisted of 86.5, 2.3, and 0.6 wt% of GalA, galactose, and arabinose, respectively. The degree of methyl esterification of pectin was 43%, Mw—401 kDa. For the adhesion test, cotton buds were immersed in a 4% pectin solution for 10 min before being incubated for 30 min in CaCl_2_, ZnCl_2_, FeCl_3_, and AlCl_3_. One hydrogel probe was incubated in a 2.0 mL microtube containing 0.4 mL of the respective salt solution. Solution concentrations were 31 mM, 62 mM, and 124 mM for CaCl_2_, ZnCl_2_, and 21 mM, 42 mM, and 84 mM for FeCl_3_ and AlCl_3_. After gelation, hydrogel probes were washed in distilled water three times. Figure 15 shows the appearance of CaP and FeP hydrogel probes. All tests were performed on freshly prepared hydrogels.

For the compression test, hydrogel samples were prepared by immersing two dialysis bags, each containing 50 mL of 4% pectin solution, for 24 h in 200 mL of CaCl_2_, ZnCl_2_, FeCl_3_, or AlCl_3_ solution. Solution concentrations were 31 mM, 62 mM, and 124 mM for CaCl_2_, ZnCl_2_, and 21 mM, 42 mM, and 84 mM for FeCl_3_ and AlCl_3_. After gelation, the hydrogel was taken out of the dialysis bags, cut into cubes (10 mm in all dimensions), and washed in distilled water three times. For the implantation study, hydrogel samples measuring 10 × 10 × 5 mm (width, length, and height) and weighing 0.2–0.4 g were cut from the hydrogel cube. Figure 16 shows the appearance of the CaP and FeP hydrogel cubes. All tests were performed on freshly prepared hydrogels.

### 4.2. Mechanical Properties

A compression test of the hydrogel was performed using the TA-XT Plus Texture Analyzer (Texture Technologies Corp., Stable Micro Systems, Godalming, UK). The hydrogel cubes (10 mm in all dimensions) were compressed at 25 °C with a 12 mm diameter (P/0.5R) cylinder probe with a pre- and post-test speed of 5.0 mm/s and a test speed of 1 mm/s. Hardness was determined as a positive peak; brittleness as the traveling distance from the start of compression and a positive peak; and adhesiveness as a negative peak. Young’s modulus was calculated using the following equation: E = (F/A)/(ΔH/H),(1)
where F is the force (N) measured during compression, A is the cross-sectional area of the hydrogel cube, and ΔH/H is the uniaxial deformation. The calculations were performed for eight replicate samples using Texture Exponent 6.1.4.0 software (Stable Micro Systems, Godalming, UK). 

Mechanical measurements were performed at room temperature on cubic specimens with a length of 10 mm and an extension rate of 1.0 mm/s according to ASTMD 695-15, with 8 replicates.

### 4.3. Scanning Electron Microscopy

Before SEM analysis, the sample was chipped using a scalpel at room temperature. The samples were chipped and fixed with carbon double-sided adhesive tape to a 25-mm aluminum specimen stub. A carbon coating with a thin film (10–20 nm) was performed. Then, metal coating with a thin film (20 nm) of chromium alloy was performed using the magnetron sputtering method. The observations were carried out using a Hitachi SU8000 field-emission scanning electron microscope (FE-SEM). Images were acquired in secondary electron mode at 2 and 5 kV accelerating voltages and at a working distance of 8–10 mm. The morphology of the samples was studied, taking into account the possible influence of metal coating on the surface. EDS-SEM studies were carried out using the Oxford Instruments X-max 80 EDS system at 30 kV accelerating voltage and a working distance of 15 mm. 

### 4.4. Stability of FeP Hydrogel

Two FeP hydrogel probes were incubated in 3 mL of Hanks’ solution (NaCl 140 mM, KCl 5 mM, CaCl_2_ 1 mM, MgSO_4_ 0.4 mM, MgCl_2_ 0.5 mM, Na_2_HPO_4_ 0.3 mM, KH_2_PO_4_ 0.4 mM, D-glucose 6 mM) at pH 7.4, 5.0, and 8.0 for 24 h at 37 °C. Each Hanks’ solution variant was supplemented with 25 mM HEPES to maintain the set pH. 

To determine the mechanical properties, the hydrogel cubes were compressed with a 25-mm-diameter (SMS P/25) cylinder probe with a pre- and post-test speed of 5.0 mm/s, a test speed of 1 mm/s, and a travel distance of 4 mm. The calculations of maximum peaks were performed for six replicate samples using Texture Exponent 6.1.4.0 software (Stable Micro Systems, Godalming, UK). 

The concentrations of GalA and iron were determined in the incubation medium after 3, 6, and 24 h of incubation, as described below.

The GalA content was determined by reaction with 3,5-dimethylphenol in the presence of concentrated sulfuric acid [34]. Briefly, 2 mL of incubation medium was centrifuged at 3000× *g* for 5 min at 4 °C (Micro 220 R Hettich Zentrifugen, Tuttlingen, Germany). The resulting supernatant (200 mL) was precipitated with a fourfold (800 mL) volume of 96% ethanol at 4 °C for 15 h and centrifuged at 10,000× *g* for 20 min at 4 °C. The precipitate was washed twice with 96% ethanol and dissolved with 400 mL of H_2_O. Concentrated H_2_SO_4_ (2 mL) was carefully added to a mixture of the washed sample (125 µL, 20–100 μg/mL) and 125 µL borate solution (solid NaOH was added to a 20% suspension of H_3_BO_3_ in water to complete dissolution); the content of the tube was mixed and heated for 40 min at 70 °C. The tube was placed in cold water to quickly cool the reaction mixture to ambient temperature, and 3,5-dimethylphenol solution (50 µL) was added. The mixture was incubated at room temperature for 10 min, and the absorbance was recorded with a Power-Wave 200 (Bio-Tek Instruments, Santa Clara, CA, USA) at 450 and 400 nm against an appropriate blank containing water instead of GalA solution. The difference in absorbances at these two wavelengths was used to calculate the GalA content. A calibration curve was built with GalA (Sigma-Aldrich, Buchs, CH, USA).

The concentration of iron in the incubation medium was determined using the Iron-Agat kit (Agat-Med, Moscow, Russia). Briefly, 33 µL of incubation medium was added to the wells of a polystyrene plate, and 167 µL of hydroxylamine working solution was added. After 30 min of incubation at room temperature, 33 μL of sodium acetate was added to the wells, mixed thoroughly, and the optical density was measured at 562 nm. Next, 1.3 μL of ferrozine solution was added to the contents of the wells, which were incubated for 5 min, and again, the optical density was measured at 562 nm.

### 4.5. Serosal Adhesion

The force of hydrogel adhesion to rat small intestine serosa was measured to evaluate the bioadhesive properties of the pectin hydrogels [32]. The prepared hydrogel probe was fixed by a grip (Stable Micro Systems, Godalming, UK) above rat serosa fixed by a film support rig (Stable Micro Systems, Godalming, UK) (Figure 17). The serosa was compressed by the hydrogel probe at various forces (25, 50, or 100 mN) for 20, 60, or 120 s in the single adhesion measuring probe. The measured compression force in repeated adhesions was 50 mN, the time of compression was 10 s, and the time between compressions was 10 s. The force of probe separation from the tissue was recorded and calculated using Exponent Stable MicroSystems (Version V6.1.4.0) (Godalming, UK). Adhesion strength was determined as the maximum positive peak, and the work of adhesion was determined as the area under the force overtime curve. To investigate the influence of the salt composition of the adhesion medium, the adhesion test was carried out in 0.1 mL of distilled water, PBS, Hanks’ solution, Hanks’ solution supplemented with 10% fetal bovine serum (FBS), and Hanks’ solution supplemented with 10% FBS and rat peritoneal leukocytes (5 × 10^6^ cells/mL).

### 4.6. Subcutaneous Implantation of Hydrogels

Wistar albino rats (250–300 g) were randomly divided into three groups of three animals each, according to the following protocol: group I (Control, sham-operated), the negative control group, was set for the time periods of implantation with empty pockets; group II (CaP), rats with CaP hydrogel implantation; and group III (FeP), rats with FeP hydrogel implantation. Rats were anesthetized by the administration of tiletamine/zolazepam (Zoletil 100, 10 mg/kg, i.m.). An aseptic technique was used throughout the experimental period. The pectic hydrogel samples measuring 10 × 10 × 5 mm (width, length, and height) and weighing 0.2–0.4 g were prepared as described above. Under surgically sterile conditions, two full-thickness skin longitudinal incisions (about 2 cm) containing the subcutis and the panniculus carnosus (skin and smooth muscle) were performed in the dorsum of each animal. The hydrogel samples were inserted into these pockets, and the panniculus carnosus and the skin were carefully sutured.

After each predetermined implantation time period (1 and 7 days), each animal was anesthetized with diethyl ether, and hydrogels were retrieved from the implantation sites of each animal. The hydrogel samples were fixed with 2.5% glutaraldehyde for 15 min, washed with PBS, and stained with rhodamine and DAPI (both from Sigma-Aldrich, Burlington, MA, USA). Then, the number of cells was visually counted on each hydrogel using a light-fluorescent microscope (Altami, Russia) equipped with a digital camera. The number of adherent cells was counted and expressed as cells/mm^2^.

The rats were bled with cardiac puncture using a heparinized syringe, and the blood was separated into two aliquots. In the first aliquot of blood, the following parameters were determined: the differential white cell percentage (neutrophils, lymphocytes, eosinophils, and monocytes), using a hemocytometer and a light microscope. The second aliquot of blood was collected and stored at 40 °C by centrifugation. TNF levels in the serum samples were determined using a sandwich ELISA according to the manufacturer’s instructions. Monoclonal affinity purified anti-rat TNF-antibodies were used as capture antibodies, and biotin-conjugated anti-rat TNF-antibodies were used as detection antibodies (PeproTech Inc., London, UK). Binding was detected using horseradish peroxidase-labeled streptavidin. The color reaction was developed by the addition of H_2_O_2_ and orthophenylenediamine. The reaction was stopped after 15 min by the addition of 0.1 mL of 2.5 mol/L hydrochloric acid, and the absorbance (OD) was read at 492 nm (PowerWave 200, BioTek Instruments, Winooski, VT, USA).

### 4.7. Statistical Analysis

The results were presented as the arithmetic mean ± standard deviation. A one-way ANOVA with Tukey’s honest significance test was applied to determine what was statistically significant. Statistical significance was defined as *p* < 0.05. 

## Figures and Tables

**Figure 1 ijms-24-01248-f001:**
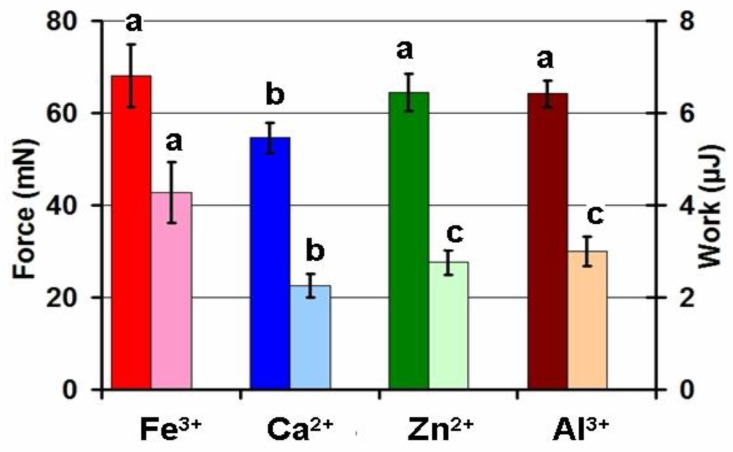
Comparison of the adhesion strength and work of adhesion of pectin hydrogels cross-linked by Fe^3+^, Ca^2+^, Zn^2+^, and Al^3+^ ions. The data are presented as the mean ± standard deviation (SD). Different lowercase letters indicate significant differences among means for the different ions (*n* = 8, *p* < 0.05). The red, blue, green, and brown bars correspond to the adhesion strengths of the hydrogels cross-linked by Fe^3+^, Ca^2+^, Zn^2+^, and Al^3+^ ions, respectively. Pink, light blue, light green, and light brown bars correspond to the work of adhesion of the hydrogels cross-linked by Fe^3+^, Ca^2+^, Zn^2+^, and Al^3+^ ions, respectively.

**Figure 2 ijms-24-01248-f002:**
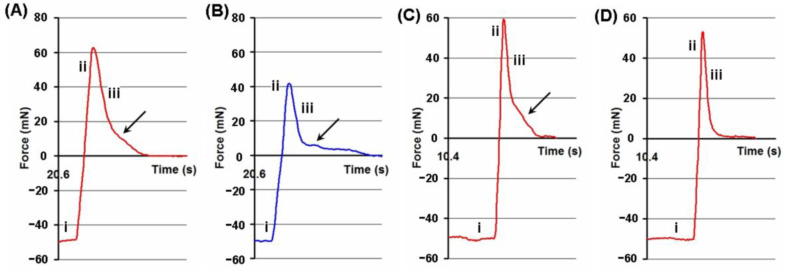
Tensile strength adhesive curve of the interaction of the pectin hydrogels with the serosa. Adhesion of FeP (**A**, red line) and CaP (**B**, blue line) hydrogels at a single attachment-detachment event with a compression force of 50 mN and a compression time of 20 s. Adhesion of FeP hydrogel at the first (**C**) and fifth (**D**) cycles of attachment-detachment with a compression force of 50 mN and a compression time of 10 s. Arrows show the protracted segment of the debonding curve. The following phases of the adhesion test are shown: the compression phase (i), the deadhesion phase (ii), and the debonding phase (iii).

**Figure 3 ijms-24-01248-f003:**
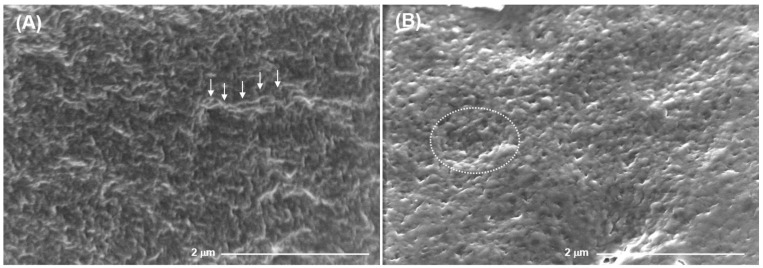
Scanning electron micrographs of the surfaces of the FeP (**A**) and CaP (**B**) hydrogels at 84 and 124 mM of Fe^3+^ and Ca^2+^, respectively. Magnification: 25,000×, scale bar: 2.0 µm. Arrows and a dotted circle indicate a representative fiber-like formation and pit, respectively.

**Figure 4 ijms-24-01248-f004:**
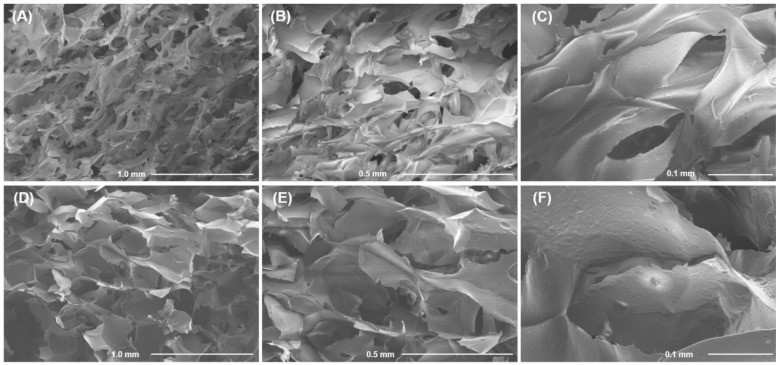
Scanning electron micrographs of the FeP (**A**–**C**) and CaP (**D**–**F**) hydrogels at 84 and 124 mM of Fe^3+^ and Ca^2+^, respectively. Magnification 50×, scale bar 1.0 mm (**A**,**D**); magnification 100×, scale bar 0.5 mm (**B**,**E**); magnification 300×, scale bar 0.1 mm (**C**,**F**).

**Figure 5 ijms-24-01248-f005:**
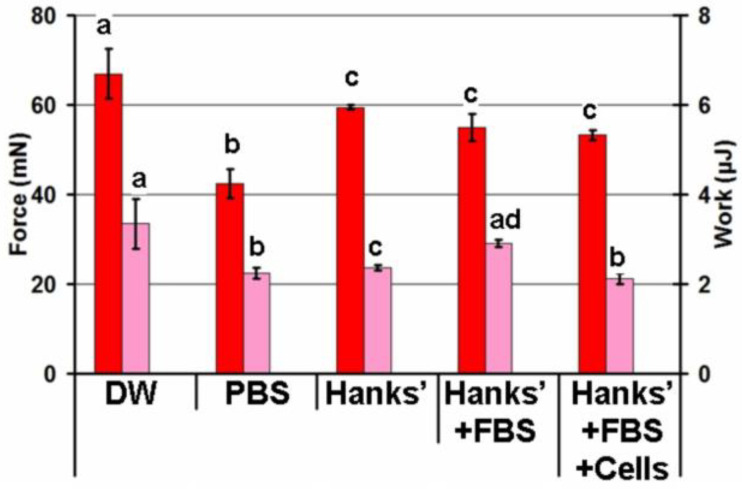
Effect of the composition of the adhesion medium on the adhesion strength and work of adhesion of the FeP hydrogel. A total of 0.1 mL of distilled water (“DW”), PBS, Hanks’ solution, Hanks’ solution supplemented with 10% fetal bovine serum (FBS), and Hanks’ solution supplemented with 10% FBS and peritoneal leukocytes (5 × 10^6^ cells/mL) were placed between the serosa and the hydrogel during the adhesion test. Different lowercase letters indicate significant differences among the means of the adhesion strength in different compositions of the adhesion medium or the means of the work of adhesion in different compositions of the adhesion medium (*n* = 8, *p* < 0.05). The red and pink bars show the adhesion strengths and work of adhesion, respectively.

**Figure 6 ijms-24-01248-f006:**
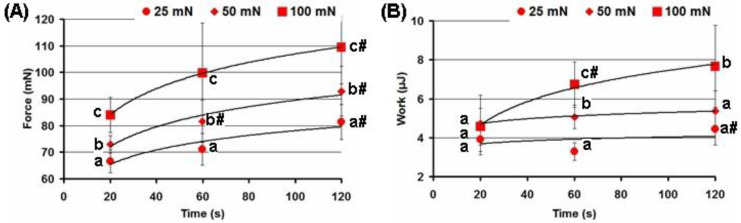
Effect of the time (20, 60, and 120 s) and force of compression (25, 50, and 100 mN) on the strength (**A**) and work of adhesion (**B**) of the FeP hydrogel to the serosa. Different lowercase letters indicate significant differences for the means of the different compression forces; #—*p* < 0.05 vs. the previous time point.

**Figure 7 ijms-24-01248-f007:**
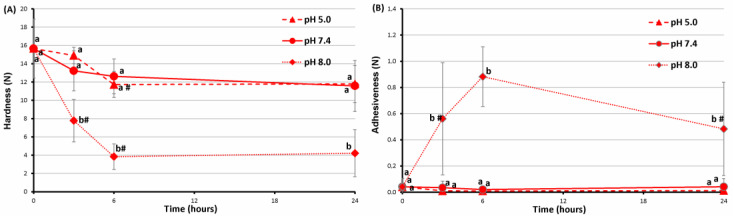
The hardness (**A**) and adhesiveness (**B**) of FeP hydrogel incubated in Hanks’ solution at a pH of 5.0, 7.4, and 8.0 for 24 h. Different letters—*p* < 0.05 for different pH conditions; #—*p* < 0.05 vs. previous time point.

**Figure 8 ijms-24-01248-f008:**
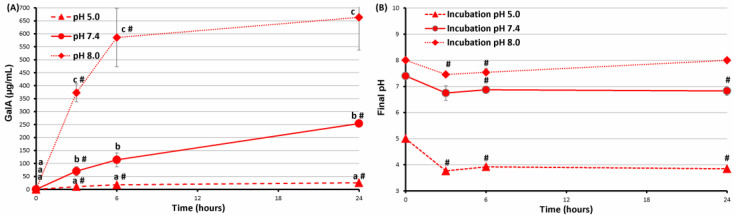
The content of GalA (**A**) and pH (**B**) of the incubation medium during incubation of the FeP hydrogel in Hanks’ solution at a pH of 5.0, 7.4, and 8.0 for 24 h. Different letters—*p* < 0.05 for different pH conditions; #—*p* < 0.05 vs. previous time point.

**Figure 9 ijms-24-01248-f009:**
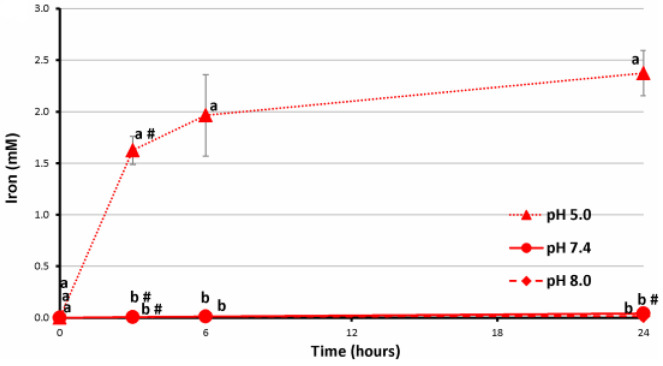
The concentration of iron ions in the incubation medium during incubation of the FeP hydrogel in Hanks’ solution at a pH of 5.0, 7.4, and 8.0 for 24 h. Different letters—*p* < 0.05 for different pH conditions; #—*p* < 0.05 vs. previous time point.

**Figure 10 ijms-24-01248-f010:**
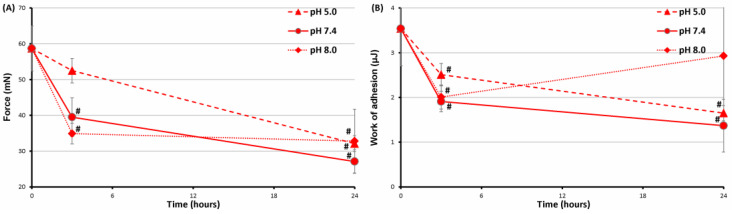
The adhesion strength (**A**) and work of adhesion (**B**) of the FeP hydrogel pre-incubated in Hanks’ solution at a pH of 5.0, 7.4, and 8.0 for 24 h. #—*p* 0.05 vs. previous time point.

**Figure 11 ijms-24-01248-f011:**
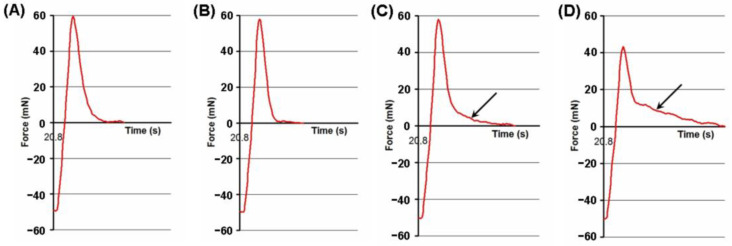
Tensile strength adhesive curve (red line) of non-treated FeP (**A**) and FeP pre-incubated in Hanks’ solution with a pH of 5.0 (**B**), 7.4 (**C**), and 8.0 (**D**) for 24 h. The black arrow indicates the protraction of the debonding curve.

**Figure 12 ijms-24-01248-f012:**
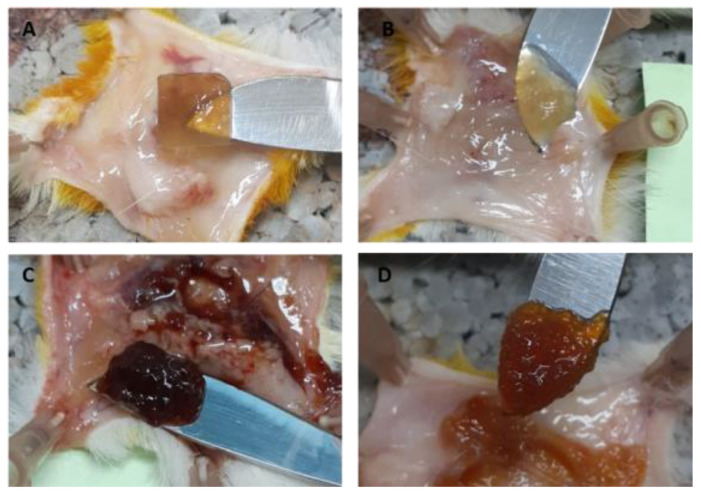
Degradation of CaP (**A**,**B**) and FeP (**C**,**D**) hydrogels during subcutaneous implantation in laboratory rats. (**A**,**C**)—after 1 day of implantation, (**B**,**D**)—after 7 days of implantation.

**Figure 13 ijms-24-01248-f013:**
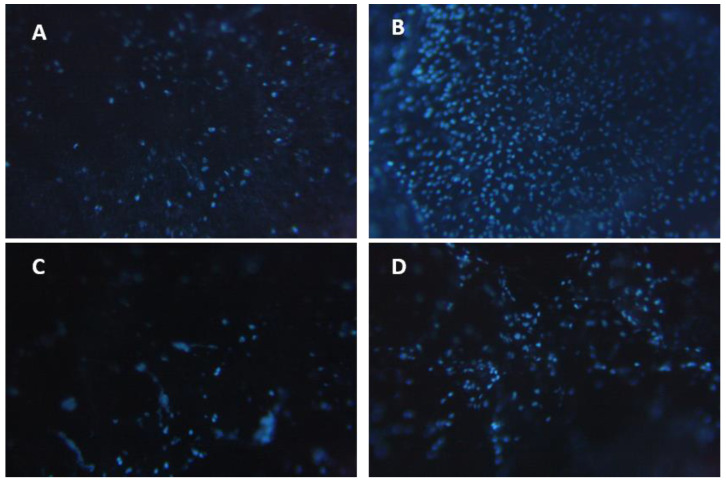
Cell infiltration of the CaP (**A**,**B**) and FeP (**C**,**D**) hydrogels during subcutaneous implantation in laboratory rats. (**A**,**C**)—after 1 day of implantation, (**B**,**D**)—after 7 days of implantation. Magnification 100×.

**Figure 14 ijms-24-01248-f014:**
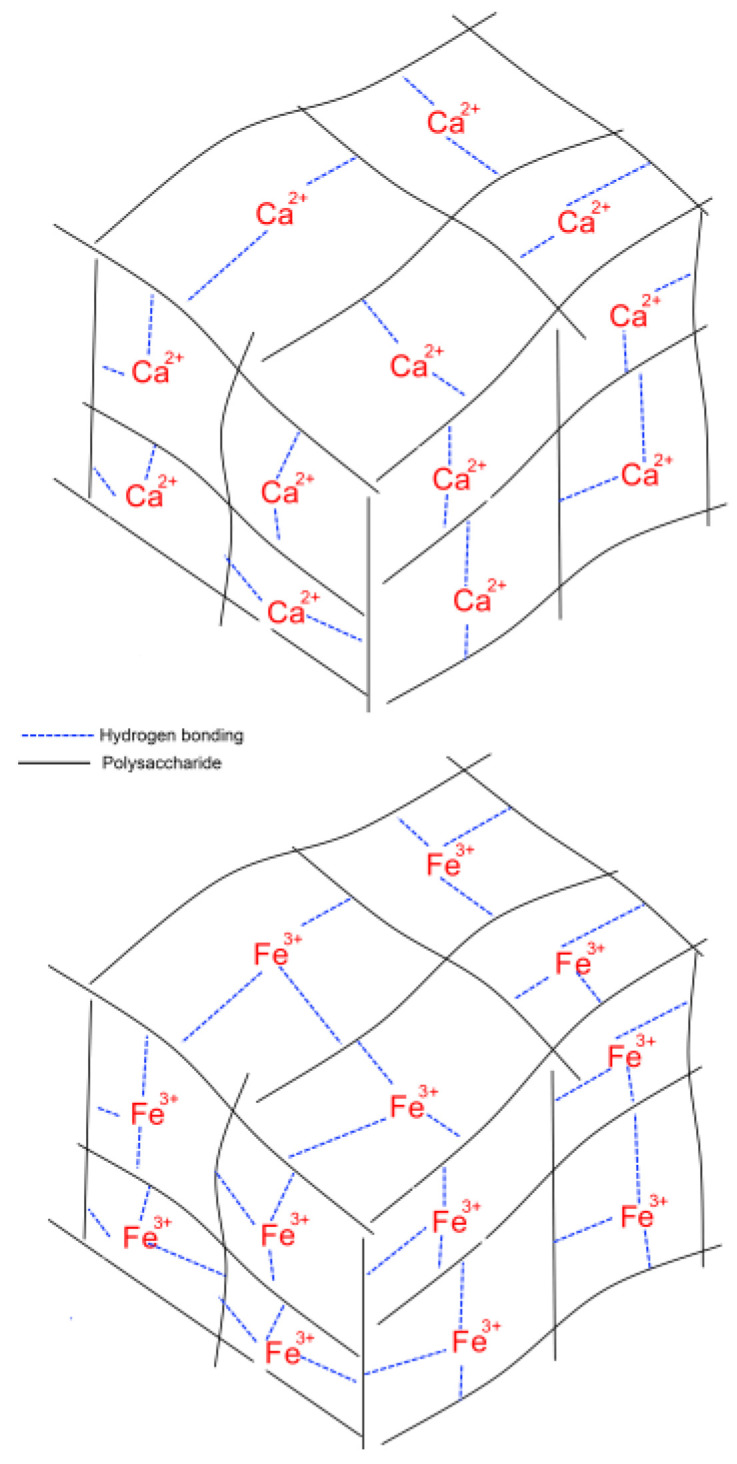
Mechanism presentation of the “egg-box” model of pectin gelation induced by Ca^2+^ and Fe^3+^ cations.

**Figure 15 ijms-24-01248-f015:**
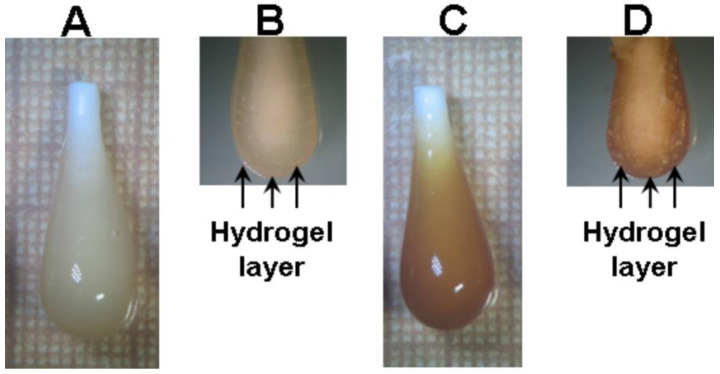
Appearance of CaP (**A**,**B**) and FeP (**C**,**D**) hydrogel probes for adhesion tests. (**B**,**D**)—the gel layer closest to the observer was cut off to see the hydrogel layer (1–2 mm).

**Figure 16 ijms-24-01248-f016:**
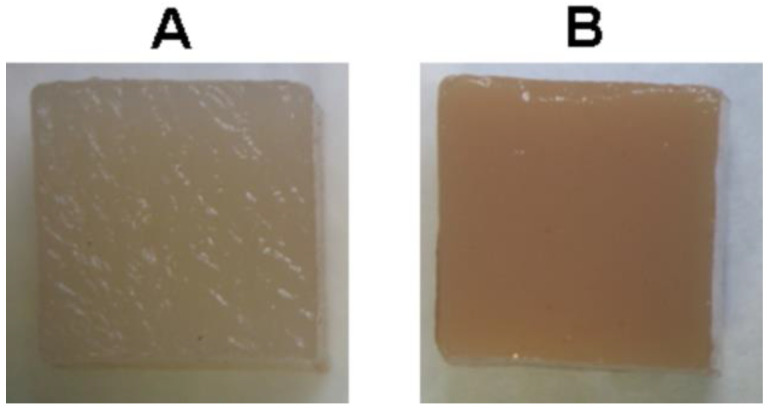
Appearance of the CaP (**A**) and FeP (**B**) hydrogel cubes for compression tests.

**Figure 17 ijms-24-01248-f017:**
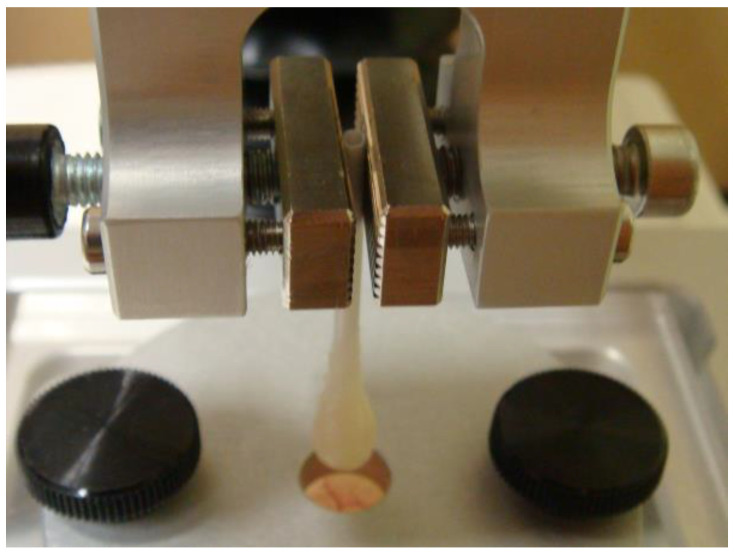
Schematic of serosa-hydrogel adhesion testing.

**Table 1 ijms-24-01248-t001:** Mechanical properties of the FeP and CaP hydrogels in a puncture test.

Hydrogel; Ion Concentration	Hardness (N)	Young Modulus (kPa)	Brittleness (mm)	Adhesiveness (N)
FeP; 21 mM	1.97 ± 0.26 ^a^#	2429 ± 1537 ^a^#	1.6 ± 0.4 ^a^	0.10 ± 0.02 ^a^#
FeP; 42 mM	2.44 ± 0.18 ^b^#	8435 ± 4115 ^b^#	0.9 ± 0.3 ^b^#	0.19 ± 0.03 ^b^#
FeP; 84 mM	1.98 ± 0.15 ^a^#	5882 ± 1824 ^b^#	1.0 ± 0.3 ^b^#	0.19 ± 0.07 ^b^#
CaP; 31 mM	0.44 ± 0.02 ^a^	1182 ± 73 ^a^	1.4 ± 0.1 ^a^	0.12 ± 0.01 ^a^
CaP; 62 mM	0.58 ± 0.02 ^b^	1319 ± 173 ^a^	1.5 ± 0.1 ^a^	0.11 ± 0.03 ^a^
CaP; 124 mM	0.67 ± 0.05 ^c^	1395 ± 281 ^a^	1.6 ± 0.2 ^b^	0.13 ± 0.02 ^a^

The data are presented as the mean ± SD. Different lowercase letters indicate significant differences among the means of different concentrations of the same cation; #—differences are significant compared to the hydrogel cross-linked by Ca^2+^ ions at the concentration of the corresponding R (*n* = 8, *p* < 0.05).

## Data Availability

The data that support the findings of this study are available from the corresponding author upon reasonable request.

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
