# Peer review of "Serosal Adhesion Ex Vivo of Hydrogels Prepared from Apple Pectin Cross-Linked with Fe3+ Ions"

_ijms, 2023, doi:10.3390/ijms24021248_

Round 1

Reviewer 1 Report

The manuscript entitled “Serosal Adhesion Ex Vivo of Hydrogels Prepared from Apple Pectin Cross-linked with Fe3+ Ions” by Sergey Popov, Nikita Paderin, Elizaveta Chistiakova, Dmitry Ptashkin, is focus on the mechanical and adhesive properties of the material, without paying attention to the physico-chemical characterization, as well as on determining its viability on specific cell lines, as would be justified in the case of studying a new biomaterial. In my opinion, the subject of the manuscript and the obtained results are scientifically relevant, but I believe that the work should be improved with additions of sections regarding the chemical structure and biological properties of the proposed biomaterial, in order to reach the scientific level of the journal.

Critical remarks are listed below:

Authors should check for typos and grammatical errors in the manuscript.

The use of the words gel and hydrogel should be consistent throughout the work.

Introduction

** Lines 34-36: please check the sentence;

** Line 42: please check the sentence;

** Lines 45-46: please check the sentence;

Results

** Line 98: please check the sentence;

** Lines 121-124: please check the sentences, it seems to be a discrepancy;

** Table 1: the explanation using lower and uppercase letters is not clear;

** Mechanical properties and surface microrelief of FeP and CaP hydrogels section should be rewritten, because the explanations are sometimes confusing; the mechanical properties study the influence of both the types of cations and their concentration, while SEM analyzes only from the point of view of the nature of the cation, so there can be no joint conclusions;

** Lines 263-265: please check the sentence;

Disscusion

**As shown, FeP adhesion to the serosa is less advantageous in PBS, so in physiological conditions;

** Lines 339-341: please check the sentence;

**An explanation of the effect of reducing serosal adhesion after 24 h incubation in Hank’s solution would be required;

** Lines 353-355:what kind of “weak chemical” bonds do you think are formed?

** Lines 362-363: please check the sentence;

Materials and methods

**Details are needed regarding the processing of the hydrogel samples, their aspect, as well as their handling and storage;

**It is necessary to specify whether you have performed the mechanical measurements in accordance with Test Standards (ASTM), if so, please provide it.

**The hydrogel cubes were dry or wet?

** What does the statement “A target-oriented approach was utilized for the optimization of the analytic measurements” mean in the SEM section?

**For the determination of GalA content, it is necessary to describe the procedure used more accurately;

**Which is relevant to the study of adhesion due to immersion in Hank’s solution for 24 h?

**Lines 413-416: there is confusion regarding the name of the compression cylinder and the samples subjected to compression;

**Please specify the adhesion testing methods used;

References:

References 15 and 25 and also 16 and 24, are the same;

References 38 and 39 are not compliant.

Reviewer 2 Report

The manuscript investigated the adhesion of the LM pectin gel crosslinked by different cations. and showed that Fe3+ crosslinked gels exhibited serosal adhesion combined with excellent stability and mechanical properties in physiological environments, which appeared to serve as a promising bioadhesive for tissue engineering. There are several concerns as follow.

1.     The pectin gel showed higher mechanical property and stability when crosslinked by Fe3+ compared to Ca2+ and other cations, does any experiment can explain the reason and mechanism?

2.     The authors believe that the morphologies of the gels affect the adhesive property greatly. What is the mechanism? Please give adequate evidence, such as contact angle, XPS, etc.

3.     It seems that the extra composition in medium decrease the adhesivity of the FeP hydrogel according to 2.3, I wonder if the cations and anions in PBS and Hanks’ solution affect the adhesion?

4.     Please specify the meaning of the letters “a, b, c” in Fig.1 and Fig.5, and the stagement of i, ii, iii in Fig.2.

Round 2

Reviewer 1 Report

The authors made many of the requested changes, so that the work was improved. However, some requirements were not understood, such as those about the chemical structure of the proposed biomaterial. The authors introduced new data on the structure of the pectin macromolecule, while they were asked to prove the chemical structure of the pectin hydrogel (the biomaterial) through physical-chemical analyses. With the regret of the lack of physical-chemical characterization of the biomaterial, I agree to publish the paper in its current form.

Reviewer 2 Report

The concerns have been well answered. The manuscript can be published as the present form.